# Paralog buffering contributes to the variable essentiality of genes in cancer cell lines

**Barbara De Kegel** [ID]**, Colm J. Ryan** [ID]*

School of Computer Science and Systems Biology Ireland, University College Dublin, Belfield, Dublin, Ireland

* colm.ryan@ucd.ie

**Data Availability Statement:** All relevant data are within the manuscript and its Supporting Information files.

**Funding:** This work was funded through an Irish Research Council (research.ie) 2017/2018

## Abstract

What makes a gene essential for cellular survival? In model organisms, such as budding yeast, systematic gene deletion studies have revealed that paralog genes are less likely to be essential than singleton genes and that this can partially be attributed to the ability of paralogs to buffer each other's loss. However, the essentiality of a gene is not a fixed property and can vary significantly across different genetic backgrounds. It is unclear to what extent paralogs contribute to this variation, as most studies have analyzed genes identified as essential in a single genetic background. Here, using gene essentiality profiles of 558 genetically heterogeneous tumor cell lines, we analyze the contribution of paralogy to variable essentiality. We find that, compared to singleton genes, paralogs are less frequently essential and that this is more evident when considering genes with multiple paralogs or with highly sequence-similar paralogs. In addition, we find that paralogs derived from whole genome duplication exhibit more variable essentiality than those derived from small-scale duplications. We provide evidence that in 13–17% of cases the variable essentiality of paralogs can be attributed to buffering relationships between paralog pairs, as evidenced by synthetic lethality. Paralog pairs derived from whole genome duplication and pairs that function in protein complexes are significantly more likely to display such synthetic lethal relationships. Overall we find that many of the observations made using a single strain of budding yeast can be extended to understand patterns of essentiality in genetically heterogeneous cancer cell lines.

## Author summary

Somewhat surprisingly, the majority of human genes can be mutated or deleted in individual cell lines without killing the cells. This observation raises a number of questions—which genes can be lost and why? Here we address these questions by analyzing data on which genes are essential for the growth of over 500 cancer cell lines. In general we find that paralog genes are essential in fewer cell lines than genes that are not paralogs. Paralogs are genes that have been duplicated at some point in evolutionary history, resulting in our genome having two copies of the same gene—a paralog pair. These paralog pairs are a potential source of redundancy, similar to a car having a spare tire. If this is the case, one might expect that losing one gene from a paralog pair could be tolerated by cells, due to

Laureate Award awarded to C.J.R. The funders had no role in study design, data collection and analysis, decision to publish, or preparation of the manuscript.

**Competing interests:** The authors have declared that no competing interests exist.

the existence of a 'backup gene', but losing both members would cause cells to die. By analyzing the cancer cell lines we estimate this to be the case for 13–17% of paralog pairs, and that this provides an explanation for why some genes are essential in some cell lines but not others.

## Introduction

Genes are classified as essential or non-essential based on whether their mutation or deletion causes cell death. Essentiality has been used as a model to understand general features of the mapping from genotype to phenotype and also to understand genetic robustness, the ability of organisms to tolerate genetic perturbations. The majority of studies of essentiality have analyzed comprehensive maps of essential genes derived from systematic genetic perturbation studies in laboratory strains of model organisms, most notably the *Saccharomyces cerevisiae* yeast gene deletion collection [1,2]. These analyses have revealed biological features that influence essentiality, for instance genes that encode protein complex subunits are more likely to be essential [3,4] while duplicate genes (paralogs) are less likely to be essential [5,6]. A limitation of these studies, imposed by a lack of data, is that essentiality is treated as a fixed binary property. However, it has become increasingly apparent that gene essentiality is influenced by genetic background effects and can be highly variable even within species [7–10]. For instance, analysis of gene deletion collections in two very closely related strains of *S. cerevisiae* revealed that ~6% of genes are 'conditionally essential', i.e. essential in only one background [11]. While such studies have revealed that essentiality is influenced by genetic background, the small number of strains analyzed have precluded a large-scale analysis of factors that contribute to the variation in essentiality.

Recent technical advances, such as the development of CRISPR-Cas9 based approaches for genome-wide screening, have made it possible to identify essential genes in large panels of cancer cell lines [12–14]. These resources make it possible, for the first time, to systematically investigate the factors that contribute to variation in essentiality. In this work we perform such a systematic analysis, focusing on the contribution of duplicate genes (paralogs) to variation in essentiality.

Gene duplication is the primary mechanism by which new genes are created. Initially gene duplication results in two genes that have identical coding sequences and typically identical gene regulatory regions. Over evolutionary time scales the two paralogs may diverge in functionality, but they frequently maintain at least some degree of functional overlap that can provide buffering capacity [15]. The relationship between gene duplication and essentiality has been well established in budding yeast and replicated in additional model organisms. Initial analysis of the yeast gene deletion collection suggested that singleton genes, i.e. those with no identifiable paralog, were more than twice as likely as paralogs to be essential [5]. Similar observations have been made using RNA interference in the worm *Caenorhabditis elegans* [6], in systematic knockout studies in mice [16], and using genetic screens of individual cancer cell lines [17]. This suggests that paralogs contribute significantly to the ability of cells and organisms to tolerate genetic perturbations, typically termed genetic robustness [5].

The simplest explanation for the increased tolerance of paralog loss is that the functional redundancy shared by paralogs allows them to buffer each other's loss. Direct evidence supporting this model comes from double perturbation screens that compare the fitness consequences of perturbing two genes simultaneously to the expected consequences based on perturbing each gene individually. The majority of these screens have been performed in *S.*

*cerevisiae*, where it has been shown that about 25–35% of paralog pairs are 'synthetic lethal', i.e. the simultaneous disruption of both paralogs in the pair is lethal, while the individual genes can be deleted without a significant effect on fitness [18–20]. In contrast, fewer than 5% of randomly selected gene pairs are synthetic lethal, suggesting that paralog pairs are significantly more likely to buffer each other's loss than random gene pairs. Subsequent analyses have clarified that various features of paralog pairs, such as their sequence similarity and mode of duplication, can influence their essentiality and propensity to buffer each other's loss [5,19,21–23]. Overall the evidence supports a model whereby paralog genes are less likely to be essential than singleton genes because the loss of one member of a paralog pair can be compensated for by the other member. However, as these analyses have been performed in a fixed genetic background there is little understanding of how genetic variation may contribute to paralog essentiality.

Here we use the results of genome-scale CRISPR-Cas9 screens of 558 genetically heterogeneous cancer cell lines [12] to address this question. We find that paralogs are less likely to be broadly essential than singleton genes and more likely to be never essential in any of the cell lines assayed. The frequency of essentiality of a given paralog can be associated with the number of paralogs it has, their sequence similarity and its mode of duplication (whole-genome vs. small-scale duplication). We estimate that in 13–17% of cases where a paralog is essential in some cell lines but not others, this variation in essentiality can be attributed to a synthetic lethal relationship with a variably expressed paralog. We find that paralogs that display such synthetic lethal relationships are enriched among whole-genome duplicates and protein complex members, and that both of these factors are independently predictive of synthetic lethality.

## Results

### Essentiality and paralogy for 16,540 human genes

To understand the variable essentiality of each gene we first obtained the results of genome-wide pooled CRISPR-Cas9 loss-of-function screens performed in 558 genetically heterogeneous cancer cell lines [12] (see Methods). These screens were performed using the Avana single guide RNA (sgRNA) library which contains sgRNAs targeting 17,634 genes with approximately 4 sgRNAs per gene. The fitness consequences of disrupting a specific gene are estimated by comparing the abundances of guides targeting the gene at the start of the experiment to the abundances of the same guides after 21 days of growth. A relative decrease in abundance of guides from the beginning to the end of the experiment suggests that disruption of the target gene causes a fitness defect. The magnitude of the impact on fitness can be inferred from the magnitude of change in abundance. By design, each sgRNA should target only a single gene, but recent work has shown that this and other libraries contain 'multi-targeting' sgRNAs that can match multiple sites in the genome and consequently disrupt the function of multiple genes [24,25]. This means that any fitness measurements made using these 'multi-targeting' sgRNAs may reflect genetic interactions resulting from inhibiting multiple genes simultaneously. Previous work has found that such multi-targeting sgRNAs disproportionately target paralog genes [24] potentially confounding any naive comparison of the fitness effects of disrupting paralogs and singletons. To avoid such confounding we reprocessed the screens to remove multi-targeting sgRNAs (Fig 1A).

We identified multi-targeting sgRNAs by performing genome alignment (see Methods) similar to previous efforts [24]. We obtained a set of paralog relationships from ENSEMBL [26] (see Methods) and as expected found that paralogs are more likely than singleton genes to be targeted by at least one multi-targeting sgRNA (Odds Ratio (OR) = 3.7, $p < 2 \times 10^{-16}$, Fisher's

**A**

Results of CRISPR knockout screens in 558 cancer cell lines

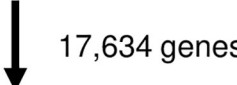
↓ 17,634 genes

**Pre-processing of gene scores**
- Drop multi-targeting sgRNAs
- Process log-fold changes with CERES
- Drop genes targeted by less than 3 guides
- Drop non-protein-coding genes

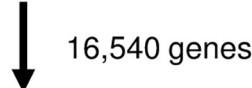
↓ 16,540 genes

**Binarize gene fitness scores**
- Call each gene in each cell line either essential or not essential

**B**

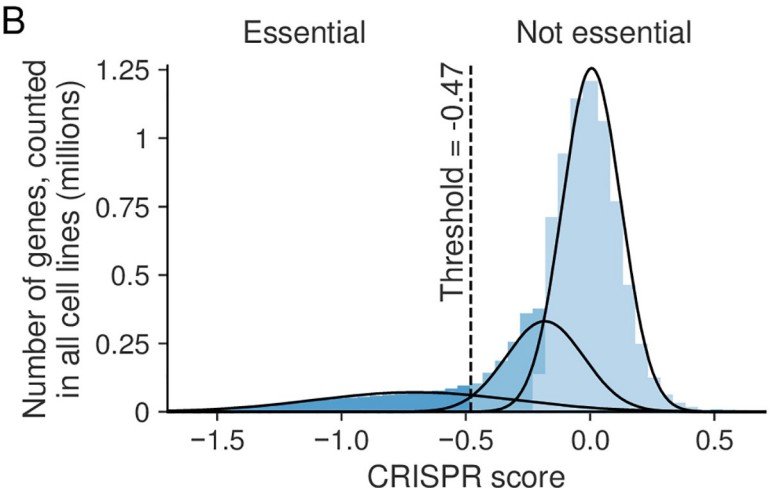

**Fig 1. Obtaining essentiality data for 16,540 protein-coding genes in 558 cancer cell lines.** (A) Workflow showing how gene scores were pre-processed and subsequently binarized. (B) Histogram showing the distribution of all gene scores across all cell lines. Curves show the 3-components of the Gaussian mixture model fit to this distribution. Colors indicate the component model each region of the histogram was assigned to (differing shades of blue for different components). The threshold (-0.47) used to convert gene scores from continuous values to binary essential and non-essential calls is shown using the dotted line.

exact test). Genes that share high sequence identity with their closest paralog were particularly vulnerable to multi-targeting; ~86% of the genes that share 90% or more sequence identity with their closest paralog are targeted by at least one multi-target sgRNA in the Avana library, compared to ~15% of genes overall (S1A Fig).

We then dropped sgRNAs that target more than one gene before reprocessing all of the CRISPR screens with CERES (see Methods) which corrects for the known artefacts in CRISPR

screens introduced by copy number variation [12,27]. CERES integrates data from all sgRNAs targeting a given gene to calculate a single fitness score per gene. However, after dropping the multi-targeting sgRNAs from the dataset, a number of genes were targeted with fewer than 3 sgRNAs. We excluded these genes from further analysis, resulting in a dataset containing fitness scores for 16,638 genes. As might be expected given their susceptibility to multi-targeting, genes with at least one highly sequence-similar paralog were disproportionately filtered out: ~67% of the genes that share 90% sequence identity or more with their closest paralog were dropped, while overall only ~8% of paralogs were dropped (S1B Fig). Moreover, most of the dropped genes are highly sequence similar paralogs (S1B Fig).

We used an established metric, precision-recall analysis, to evaluate the ability of our reprocessed scores to identify a set of common essential genes [28] compared to the gene scores in the DepMap portal (see Methods). The mean area under the precision-recall curve (AUC) for our reprocessed scores (~0.962) is marginally higher than the average AUC for the published DepMap gene scores (~0.959), indicating a slight improvement in quality. Finally, we dropped a small number of genes which were not classified as protein-coding in the HGNC [29], resulting in a final set of 16,540 genes (Fig 1A), of which 10,130 (~61%) are paralogs and the remaining 6,410 (~39%) are singletons (S1 Table).

We were interested in essentiality, the absolute requirement for a given gene in a given context, but the CERES pipeline provides quantitative scores for the fitness effects of disrupting each gene in each cell line, rather than binary classifications of essentiality. To binarize these fitness scores we used Gaussian mixture models to model the distribution of all scores across all cell lines (see Methods). We found that the distribution of fitness scores was best modelled by a mixture of three distributions, and observed that these correspond approximately to three phenotypes that result from gene disruptions: severe fitness defects, moderate fitness defects, and no obvious fitness consequences (Fig 1B). We treated gene fitness scores assigned to the first category as essential genes, and all others as non-essential, and used this binarized data for all further analyses (S2 Table). We note that similar distributions of fitness have been observed in yeast, where gene deletions can be grouped into those that are lethal, those that cause a fitness defect, and those that cause no obvious growth defect [2].

## Most genes are never or rarely essential

Before comparing singletons and paralogs, we first looked at the broad patterns of gene essentiality for all 16,540 protein-coding genes across the 558 cell lines. The median number of essential genes per cell line is 1,678, which represents ~10% of the genes in our dataset, but the number of essential genes varies widely across cell lines (S1C Fig). This is broadly in line with a recent orthogonal analysis of CRISPR screens in cancer cell lines that reported a median of 1,413/ 18,009 (~7.8%) essential genes per cell line [14] and a summary of several independent genome-wide screens of human cell lines that suggested, on average, ~11% of genes are essential [7].

A large proportion of genes—7,865 or ~48%—are never essential, that is, they can be knocked out without a severe impact on growth in any of the 558 screened cell lines (Fig 2A, blue). This number increases to 11,961, or ~72%, when including genes that are classified as essential in 1% or less of all cell lines. Only ~9% of the genes in our dataset are essential in at least half of the cell lines, while only ~6% are essential in at least 90% of the cell lines (Fig 2A, orange). Consistent with expectations, we found that these broadly essential genes (those that are essential in at least 90% of cell lines) are enriched among members of the spliceosome ($p < 2 \times 10^{-16}$, see Methods), the ribosome ($p < 2 \times 10^{-16}$), the 26S proteasome ($p < 2 \times 10^{-16}$) and the 20S proteasome ($p < 1 \times 10^{-12}$). In summary—most genes are never or rarely essential, and only a handful of 'housekeeping' genes appear to be broadly essential across all cell lines.

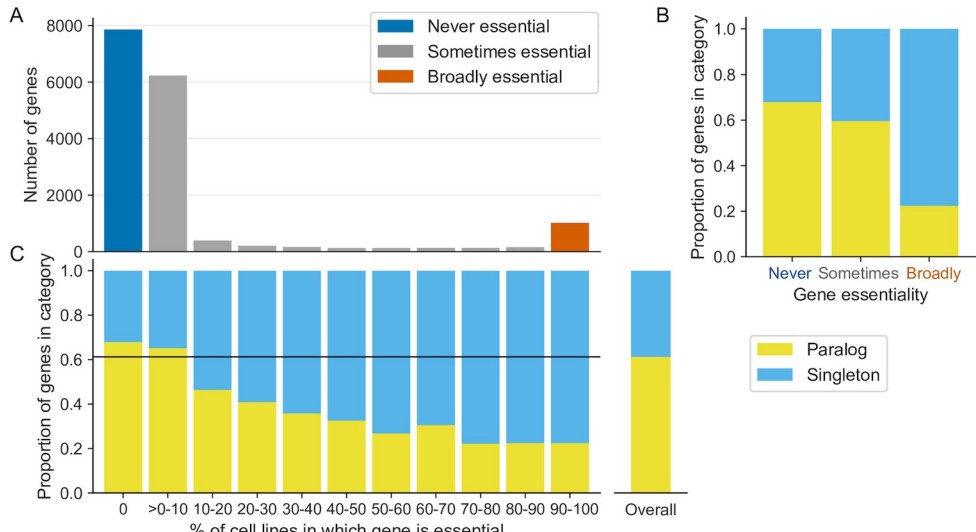

**Fig 2. Genes that are never or rarely essential are enriched in paralogs.** (A) Distribution of the percentage of cell lines in which a gene is essential. The colors indicate the three broad categories of essentiality: genes that are essential in 0% of cell lines (blue), genes that are essential in more than 0 but at most 90% of cell lines (gray), and genes that are essential in 90% or more of the cell lines (orange). (B) Stacked bar graph showing the proportion of genes in each of the three essentiality categories that are paralogs (yellow) vs. those that are singletons (cyan). (C) Stacked bar graph showing, for all genes binned according to the percentage of cell lines in which they are essential, the proportion of genes in each bin that are paralogs (yellow), or singletons (cyan). For reference, on the right is a bar showing the proportion of genes in the full dataset that are paralogs or singletons.

## Genes that are sometimes or never essential are enriched in paralogs

As noted, almost half of protein-coding genes are identified as essential in zero percent of the cell lines screened. Even after discounting these 'never' essential genes, percentage essentiality is not a normally distributed trait (Fig 2A) with a notable increase in the number of genes that are essential in at least 90% of cell lines. Consequently standard summary statistics are not appropriate to summarize the percentage essentiality and instead we partitioned the full gene set into three categories of essentiality to facilitate further analysis. These are: genes that are never essential, i.e. in 0% of the screened cell lines; genes that are sometimes essential, i.e. in at least one and less than 90% of cell lines; and genes that are broadly essential, i.e. in at least 90% of the cell lines (Fig 2A, categories shown in blue, gray and orange respectively). We consider these three categories to be representative of the different levels of gene essentiality for the purpose of cross-category comparison and refer to them throughout this work. We note that although genes are frequently classified in a binary fashion as either essential or non-essential the sometimes essential category contains ~46% of the genes analyzed (7,661/16,540).

Paralog buffering provides a plausible explanation for why some genes are never essential or only sometimes essential. If paralogs buffer gene loss, we would expect genes that are never or sometimes essential to be classified as paralogs more frequently than broadly essential genes. Indeed, we found that the never essential genes are significantly more likely to have at least one paralog compared to genes that are essential in one or more cell lines (Fig 2B, OR = 1.7, $p < 2 \times 10^{-16}$, Fisher's exact test). In contrast, broadly essential genes, i.e. genes that are essential in the vast majority of cell lines, are significantly more likely to be singleton genes (Fig 2B, OR = 6.1, $p < 2 \times 10^{-16}$, Fisher's exact test).

To see how the relationship between paralogy and essentiality varied between the two extremes of never and broadly essential we binned all genes that were essential in at least one

cell line into 10 different groups (>0–10% essential, 10–20% essential and so on). Overall, we observed an inverse correlation between the essentiality percentage and the proportion of paralogs in each bin (Fig 2C). This demonstrates not only that genes with paralogs are generally less essential than singletons but also that there is a relationship between the number of cell lines in which a gene is essential and the probability that that gene has at least one paralog. The first observation is in accordance with findings from *S. cerevisiae* and small-scale screens of human genes [7,30], whereas the latter observation can only be made based on gene essentiality data for a large number of cell lines, and was not, to our knowledge, previously reported.

## Paralog gene essentiality is influenced by the number of paralogs and their sequence identity

As a group, paralogs are less frequently essential than singleton genes, but there is still considerable variation in the frequency of essentiality between different paralogs. We anticipated that genes that have more paralogs would be less frequently essential, as there is more potential that at least one of their paralogs could compensate for their loss. To test this hypothesis we analyzed the relationship between the number of paralogs a gene has and the frequency of its essentiality. Of the genes with at least one paralog ~32% have a single paralog while the remainder have multiple paralogs (Fig 3A). We found that, as the number of paralogs a gene has increases (from one to four or more), the probability of the gene being never essential increases and the probability of the gene being broadly essential decreases (Fig 3B, $p<2\text{x}10^{-16}$, chi-squared test) indicating that essentiality is not independent of the number of paralogs. This result suggests that genes with more paralogs are essential in fewer genetic contexts.

An alternative explanation for the variation in paralog essentiality is that more similar paralogs are better able to compensate for each other's loss. Although there are many ways to measure the functional similarity of a paralog pair, we focused on protein sequence identity. To test the hypothesis that sequence identity between paralogs influences their essentiality, we annotated each paralog with the sequence identity of its closest paralog. We found that the sequence identity of the closest paralog was significantly lower for broadly essential paralogs compared to sometimes or never essential paralogs (median sequence identity of ~42%, ~47% and ~48% respectively)(Fig 3C, $p = 0.004$, Mann-Whitney U test). In contrast, there was no significant difference between the sequence identity of paralogs that were never and sometimes essential ($p = 0.5$, Mann-Whitney U test). This suggests that paralogs which are essential in fewer genetic contexts have a higher percentage of their protein sequence matched in (one of) their paralog(s).

## Whole genome duplicates display more variable essentiality than small-scale duplicates

In budding yeast, analysis of the yeast gene deletion collection has shown that paralogs that arose through different gene duplication mechanisms vary in terms of their essentiality [21,22]. Paralogs arising from whole genome duplication events are less likely to be essential than paralogs arising from small-scale duplications, while both types are less likely to be essential than singleton genes. To understand whether similar observations could be made regarding the variability of essentiality in tumor cell lines, we labelled all paralogs in our dataset as either a whole genome duplicate (WGD) or a small-scale duplicate (SSD) based on data from [31,32] (see Methods, S3 Table). Of the 10,130 paralog genes in our dataset, ~64% were annotated as WGDs while ~36% were annotated as SSDs.

We split our paralogs into two groups (WGDs and SSDs) and compared them to singleton genes. As expected, paralogs of either type were more likely than singleton genes to be never essential (OR = 1.6 and $p<2\text{x}10^{-16}$ for WGDs, OR = 1.9 and $p<2\text{x}10^{-16}$ for SSDs, Fisher's exact

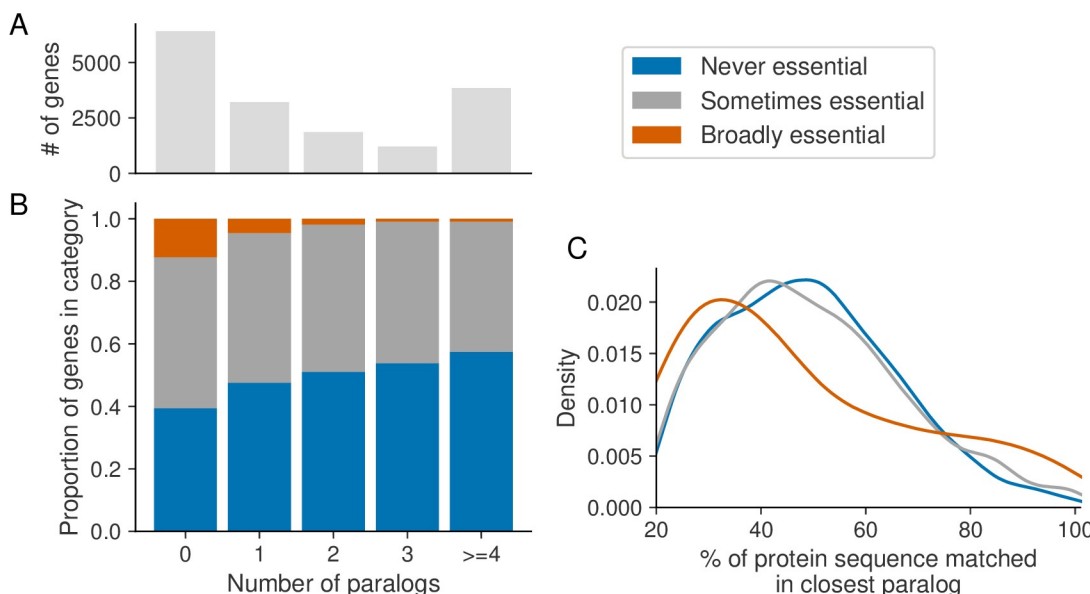

**Fig 3. Genes with more paralogs and genes which share greater sequence identity with their closest paralog are less frequently essential.** (A) Bar chart showing counts of the number of paralogs, from zero to four or more, for all protein-coding genes. (B) Stacked bar graph showing, for genes binned according to the number of paralogs they have, the proportion of genes in each bin that are never, sometimes and broadly essential. (C) The kernel density estimates of the percent of a gene's protein sequence that is identical in its closest paralog, for paralogs in each essentiality category. Drawn using Seaborn's kdeplot function with the default parameters.

test) and less likely to be broadly essential (OR = 0.1 and $p$<2x10$^{-16}$ for WGDs, OR = 0.3 and $p$<2x10$^{-16}$ for SSDs, Fisher's exact test). When comparing the two paralog groups (WGDs vs SSDs) directly we noted an interesting trend (Fig 4A). Compared to SSDs, WGDs were less

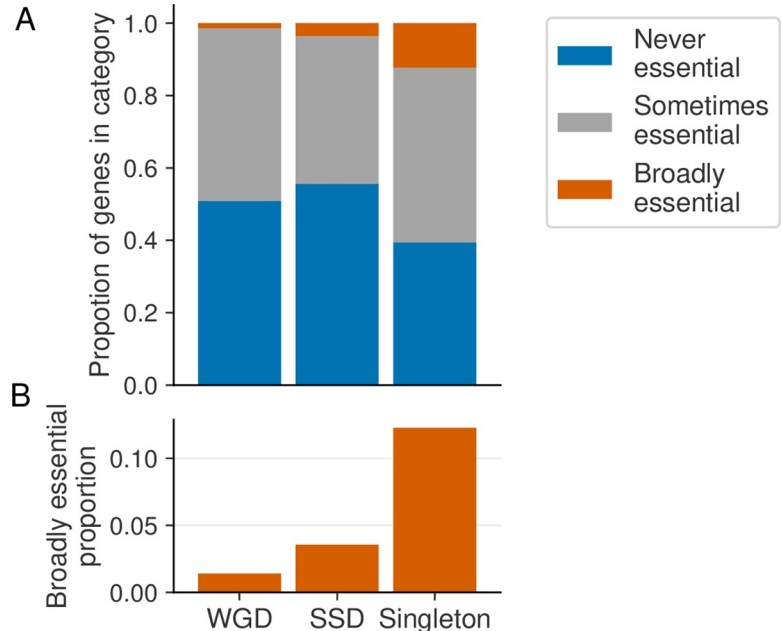

**Fig 4. WGDs are more variably essential than SSDs.** (A) Stacked bar graph showing the proportions of WGDs, SSDs and singletons that are never, sometimes and broadly essential. (B) Bar graph showing the proportion of WGDs, SSDs and singletons that are broadly essential (zoomed in version of the orange portion in A).

likely to be broadly essential ([Fig 4B], OR = 0.4, $p < 1 \times 10^{-13}$, Fisher's exact test), but more likely to be sometimes essential (OR = 1.3, $p < 1 \times 10^{-8}$, Fisher's exact test). WGDs were also less likely to be never essential (OR = 0.9, $p < 1 \times 10^{-3}$, Fisher's exact test), This suggests that WGDs are more variably essential than SSDs; they are less likely to be ubiquitously essential but more likely to be essential in a subset of cell lines.

Having observed that several features of paralogs influence their essentiality, we asked whether the combination of these features is more predictive of paralog essentiality than any of the features alone. To answer this question we built four logistic regression models to predict the essentiality category (never, sometimes or broadly essential) of paralogs. One model predicts using all three features (number of paralogs a gene has, the sequence identity of its closest paralog, and its mode of duplication) while the others each predict using only a single feature. We found that the full model, which incorporates all three features, is a significantly better fit for the data than any of the single-feature models ($p < 2 \times 10^{-16}$, likelihood ratio test, for the full model vs. each single-feature model). This indicates that the combination of features explains the variation in paralog essentiality better than any of the individual features alone.

## Paralog buffering with variably expressed genes contributes to the variable essentiality of paralogs

In budding yeast systematic studies have shown that many paralog pairs exhibit synthetic lethal or synthetic sick relationships—loss of one member of a paralog pair is associated with increased sensitivity to the inhibition of the other member [18,19]. This provides direct evidence of buffering between the two paralogs as loss of one can be tolerated but loss of both together causes a fitness defect. As the tumor cell lines we analyzed exhibit considerable variation in gene expression we hypothesized that such buffering relationships, combined with variable paralog expression, could contribute to the variable essentiality of paralogs. In these instances genetic or epigenetic variation between cell lines could result in the reduced expression of one member of a paralog pair in a subset of cell lines, rendering those cell lines especially sensitive to the inhibition of the other paralog ([Fig 5A]).

To test this hypothesis we analyzed paralogs that were sometimes, but not broadly, essential. To maintain statistical power we restricted our analysis to genes that are essential in at least 1% of cell lines. In addition, we required these genes to have at least one paralog for which we had expression data (see [Methods]). This resulted in 1,819 sometimes essential paralogs to test. We asked for each of these genes how often there was a difference in the mean expression of their most sequence-similar paralog when comparing cell lines where the gene was essential to cell lines where the gene was not essential ([Fig 5A]). For clarity we refer to the sometimes essential gene as A1 and its most sequence-similar paralog as A2. Of the 1,819 pairs tested, we found 238 pairs for which the mean expression of A2 is significantly lower in the cell lines where A1 is essential compared to the cell lines where A1 is not essential ([Fig 5B], [S4 Table], t-test FDR<10%). Thus for ~13% of the paralogs with variable essentiality, their essentiality can be associated with lower expression of their most sequence-similar paralog, suggesting a potential synthetic lethal relationship. These pairs include many of the synthetic lethal relationships between paralogs previously reported in the literature, including *ARID1A/ARID1B* [33], *SMARCA2/SMARCA4* [34,35], *STAG1/STAG2* [36,37], *ENO1/ENO2* [38], and *RPL22/RPL22L1* [39] ([Fig 5C]).

We focused our analysis on gene expression data because many different types of genetic and epigenetic variation ultimately result in changes to mRNA abundance. This can include direct (*cis*) variation of a gene or its regulatory region, e.g. hypermethylation of promoter regions often results in reduced expression of associated mRNAs, copy number amplification

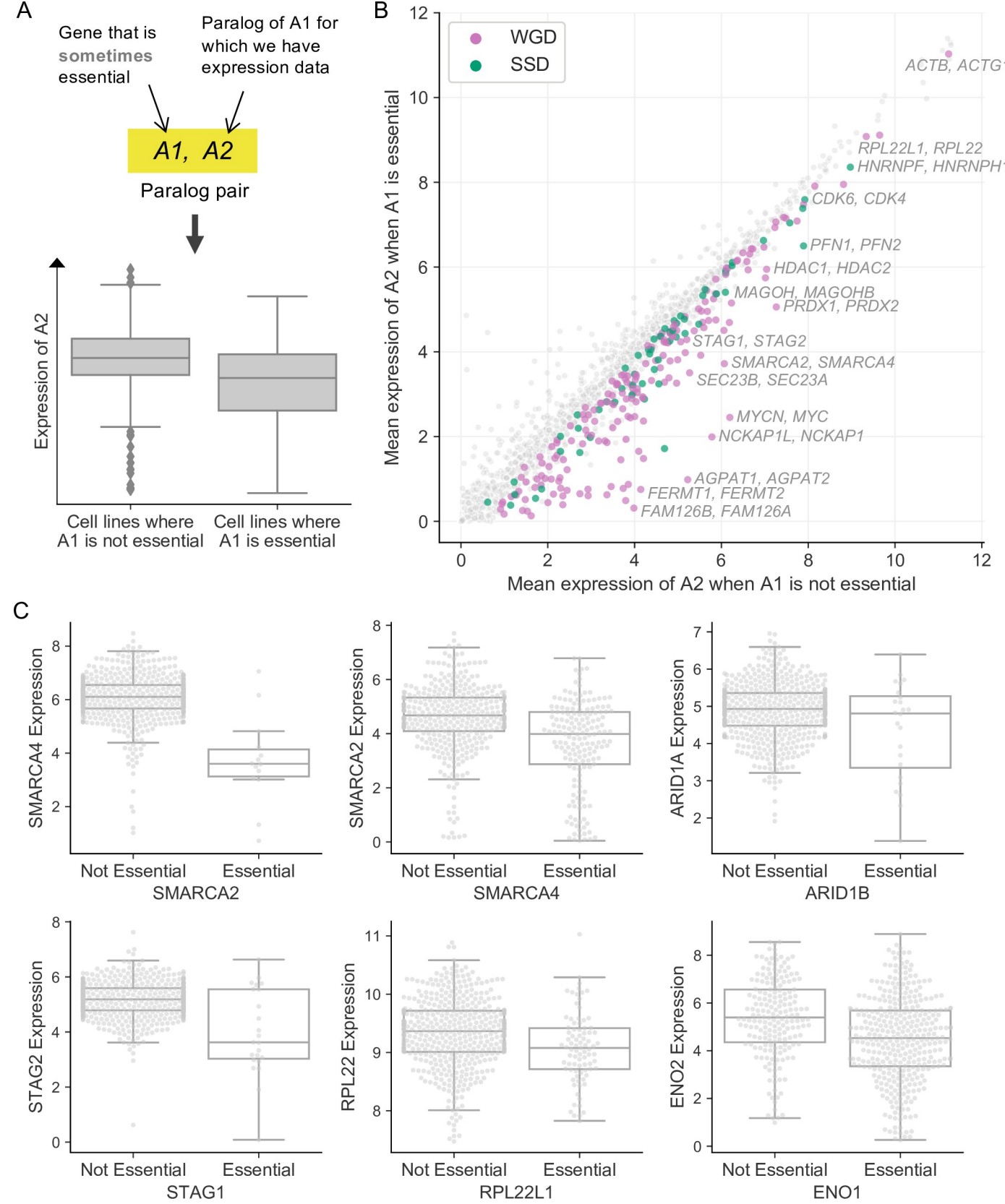

**Fig 5. Variation in paralog expression explains why some paralogs are variably essential.** (A) Top: Workflow showing selection of paralog pairs (A1, A2). Bottom: Boxplot illustrating the hypothesis that was tested for each of these pairs, namely that the expression of A2 would be higher (on average) in the cell lines where A1 is not essential compared to the cell lines where A1 is essential. (B) Scatterplot of all tested pairs showing the mean expression of A2 in the cell lines where A1 is not essential (x-axis) or essential (y-axis). Colored dots indicate pairs for which the expression of A2 is significantly lower in the cell lines where A1 is essential, i.e. putative synthetic lethal pairs. Grey dots represent pairs that were tested but for which no significant difference in expression was found between cell lines where A1 is and is not essential. Color corresponds to duplication type—pink for WGDs and green for SSDs. Selected synthetic lethalities involving protein complex subunits are labelled. (C) Boxplots showing gene pairs (e.g. A1 = *SMARCA4*, A2 = *SMARCA2*) for which A2 expression is on average higher in the cell lines where A1 is not essential compared to cell lines where A1 is essential.

of a gene often results in increased abundance of associated mRNA, while nonsense DNA mutations often result in reduced mRNA abundance due to nonsense-mediated decay. Variation in mRNA expression can also capture indirect *trans* effects, e.g. deletion of a transcription factor resulting in reduced expression of genes located on different chromosomes. To identify how many of the 238 putative synthetic lethal effects were driven by genetic changes to the paralog, rather than epigenetic or trans effects, we integrated both copy number profiling and exome sequencing data. We first asked how often the putative synthetic lethal effects were likely driven by copy number variation. For 85 of the 238 putative synthetic lethal pairs the copy number of A2 is also significantly lower in the cell lines where A1 is essential compared to the cell lines where A1 is not essential ($p<0.05$, FDR$<10\%$, t-test). We next asked how many of the putative synthetic lethal effects could be attributed to nonsense mutations. Among the 238 putative synthetic lethal pairs, there are 92 pairs where the A2 gene is subject to a nonsense mutation in at least one of the screened cell lines. For 3 of these 92 pairs there is an enrichment of nonsense mutations in A2 in the cell lines where A1 is essential compared to the cell lines where A1 is not essential ($p<0.05$, FDR$<10\%$, Fisher's exact test). All three of these pairs (*ARID1A*/*ARID1B*, *SMARCA2*/*SMARCA4* and *STAG1*/*STAG2*) involve tumor suppressor genes and have previously been reported in the literature [33,34,36,37].

## Whole genome duplicates and protein complex subunits are more likely to display putative synthetic lethality

We had already noted that WGDs are more variable in their essentiality than SSDs. A potential explanation for this increased variability is that WGDs are more likely to buffer each other's loss than SSDs. Indeed, in yeast it has been shown that WGD pairs are more likely to be synthetic lethal than SSD pairs [22]. To test if this is the case across the panel of cancer cell lines, we analyzed the set of 1,819 paralog pairs, of which we identified 238 as putative synthetic lethal using transcriptomics, to see if they were enriched in paralogs derived from a specific duplication mode. We found that ~16.5% of WGD pairs in this set are putative synthetic lethal, while only ~7.9% of the SSD pairs in the set are putative synthetic lethal (Fig 6A, OR = 2.3, $p<1\mathrm{x}10^{-7}$, Fisher's exact test). This suggests that WGD paralogs are significantly more likely to buffer each other's loss than SSD pairs.

We noted that many of the putative synthetic lethal interaction pairs identified involved genes known to encode protein complex subunits, including the majority of the pairs known from the literature. For example *ARID1A/ARID1B* and *SMARCA2/SMARCA4* both function as part of the SWI/SNF protein complex [40] while *STAG1/STAG2* function as part of the cohesin complex [36]. This suggests that paralogs coding for protein complex subunits may be more likely than other genes to show evidence of a synthetic lethal relationship. To test this hypothesis systematically, we integrated the set of 1,819 paralog pairs with a set of manually curated protein complexes [41], and found that there was a significant overlap between putative synthetic lethality and protein complex membership. Approximately 19.1% of the paralog pairs that consist of at least one protein complex member gene are putative synthetic lethal,

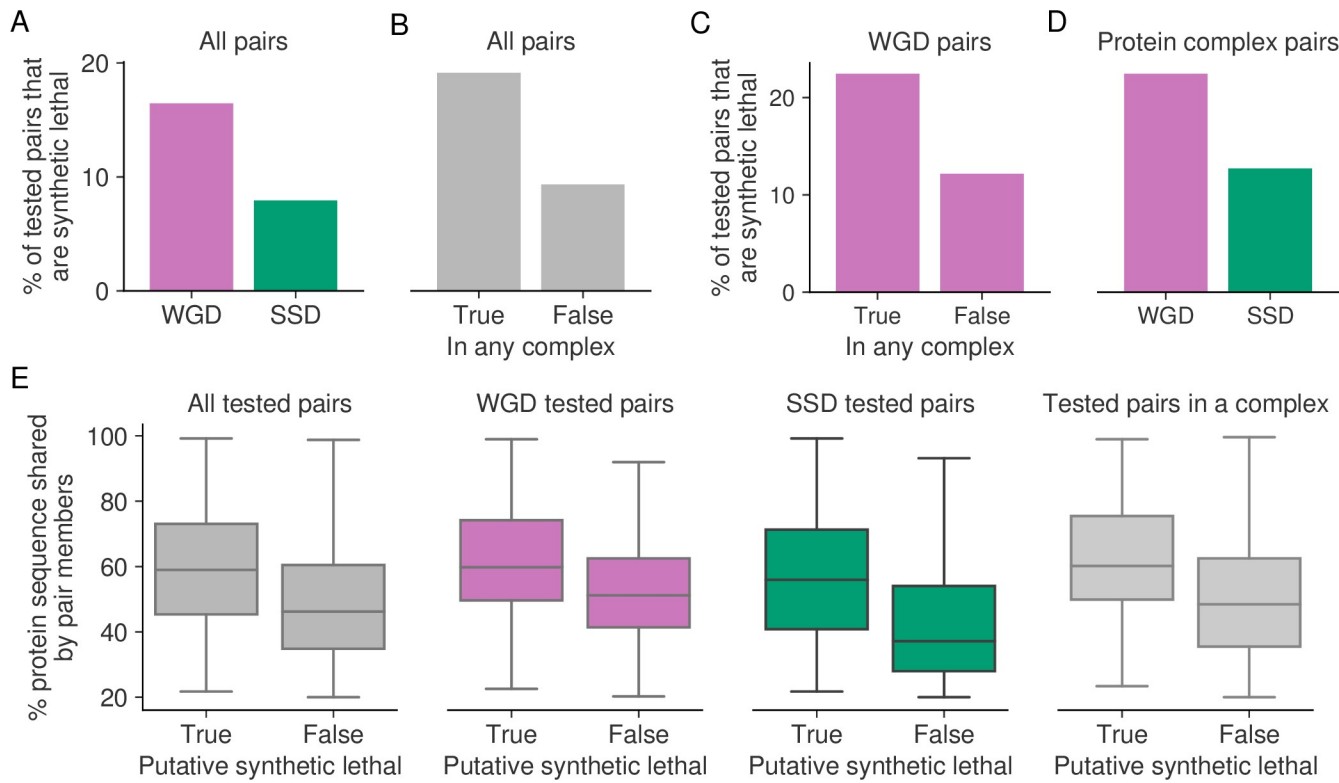

**Fig 6. Duplication mode, protein complex membership and sequence similarity are predictive of synthetic lethality.** (A) Bar chart showing the percentage of tested WGD and SSD pairs that are synthetic lethal. (B) Bar chart showing the percentage of synthetic lethal pairs observed among the tested pairs where at least one gene is a protein complex member versus the tested pairs where neither gene is a protein complex member. (C) Same as (B) but restricted to tested pairs that are WGDs. (D) Same as (A) but restricted to tested pairs where at least one gene is a protein complex member. (E) Boxplots showing the percent of the protein sequence that is identical in the members of a given paralog pair, for pairs that are synthetic lethal versus those that are not. The pairs used in the analysis consist of all sometimes essential paralogs and their most sequence-similar paralog, and three subsets thereof. From left to right the subsets consist of: pairs that are WGDs, pairs that are SSDs, and pairs where either gene is a protein complex member.

compared to only ~9.3% of the non-protein-complex pairs (Fig 6B, OR = 2.3, $p<1\times10^{-8}$, Fisher's exact test). In addition, of the pairs where both paralogs are members of the same protein complex, ~26.2% are putative synthetic lethal, compared to ~12.1% of pairs where this is not the case (OR = 2.6, $p<1\times10^{-4}$, Fisher's exact test).

A notable difference between whole genome and small-scale duplications is that the former preserves the relative gene dosage balance while the latter does not. The dosage-balance hypothesis predicts that changes in relative dosage are most likely to be deleterious for genes whose products interact in a stoichiometric fashion, such as in a protein complex [42]. Consequently small-scale duplications of protein complex subunits, which disrupt protein complex stoichiometry, will not be well tolerated while whole genome duplications, which duplicate entire complexes and maintain stoichiometry, will be better tolerated [31]. Over time this will result in a relative increase in the number of WGDs in protein complexes compared to SSDs. Indeed, in the full set of paralog pairs, we found that genes which form part of a WGD pair are significantly more likely to be protein complex members than genes in SSD pairs (OR = 2.3, $p<2\times10^{-16}$, Fisher's exact test, S2A Fig), as well as significantly more likely to be members of the same protein complex (OR = 5, $p<2\times10^{-16}$, Fisher's exact test), which is in agreement with previous observations in yeast [21].

This suggests that one reason protein complex subunits are more likely to be involved in a putative synthetic lethal interaction is that they are enriched in WGD pairs and vice versa. To

test whether the two factors were independent we first asked whether protein complex membership is still predictive of synthetic lethality when only considering WGDs. We found that WGD protein complex members are more likely to be potentially synthetic lethal than WGDs which are not protein complex members, both when considering either gene being a complex member (Fig 6C, OR = 2.1, $p < 1 \times 10^{-5}$, Fisher's exact test) or both genes being members of the same complex (OR = 1.9, $p = 0.015$, Fisher's exact test). We next asked whether, among the paralog pairs involving protein complex subunits, WGDs were enriched in putative synthetic lethals compared to SSDs. We found this to be the case: ~22.4% of WGDs in this subset are putative synthetic lethal compared to ~12.7% of SSDs in this subset (Fig 6D, OR = 2, $p = 0.002$, Fisher's exact test). This suggests that both the mode of duplication and protein complex membership are independently predictive of whether a paralog pair will exhibit evidence of a buffering relationship.

The above approach to identify putative synthetic lethal interactions was performed using only the most sequence-similar paralog for each gene. However, these paralog pairs still encompass a range of different sequence similarities. In yeast it has been shown that buffering paralog pairs share higher sequence similarity than non-buffering paralog pairs [18,43]. To test whether that is also the case among paralog pairs in cancer cell lines, we compared the sequence identities of the 238 putative synthetic lethal pairs to the sequence identities of the other sometimes essential pairs that we tested. We found that indeed, the sequence identity shared by putative synthetic lethal pairs is on average higher than that shared by non-synthetic-lethal pairs (Fig 6E, $p < 1 \times 10^{-16}$, $t$-test). The same observation can be made for different subsets of the tested pairs (Fig 6E), i.e. when only considering WGDs ($p < 1 \times 10^{-9}$, $t$-test), SSDs ($p < 1 \times 10^{-5}$, $t$-test) or pairs for which either gene is a protein complex member ($p < 1 \times 10^{-9}$, $t$-test). This indicates that more sequence-similar paralog pairs are better able to buffer each other's loss, potentially due to their greater functional overlap, and that sequence similarity is another factor that is independently predictive of synthetic lethality.

A given gene may have multiple paralogs and any one of these paralogs may be able to buffer the gene's loss. It is not necessarily the case that the most sequence-similar paralog will be responsible for this buffering relationship and consequently we may miss additional buffering relationships by looking solely at the most sequence-similar pairs. We therefore repeated our analysis of differential paralog expression using all paralog pairs where A1 is sometimes essential, and A2 has associated expression data. There are 6,221 such pairs, comprising 1,819 unique A1 genes. At a 10% FDR we found that for 374 of these pairs the expression of A2 is significantly lower in the cell lines where A1 is essential compared to those where A1 is not essential (S2B Fig, S5 Table); 211 of the 374 pairs consist of the most sequence-similar paralog for the given A1. In some cases the essentiality of a given A1 gene can be associated with reduced expression of multiple of its A2 paralogs. The 374 putative synthetic lethal pairs identified in this analysis comprise 301 unique A1s, or ~16.5% of the tested sometimes essential paralogs, that have their essentiality associated with lower expression of one or more of their paralogs. We again asked how often this putative synthetic lethal effect was likely driven by copy number variation or genetic mutation. For 110 of the 374 putative synthetic lethal pairs the copy number of A2 is significantly lower in the cell lines where A1 is essential compared to the cell lines where A1 is not essential ($t$-test, $p < 0.05$, FDR < 10%). Analyzing nonsense mutations we found only the same three pairs as in the previous analysis (*ARID1A/ARID1B*, *SMARCA2/SMARCA4*, *STAG1/STAG2*) (Fisher's exact test, $p < 0.05$, FDR < 10%).

As with the analysis restricted to the most sequence-similar paralogs only, we found that the putative synthetic lethal pairs in this set of expanded paralog pairs are enriched for WGDs compared to SSDs (S2C Fig, OR = 3, $p < 2 \times 10^{-16}$, Fisher's exact test), and for protein complex membership (S2D Fig, OR = 2.2, $p < 1 \times 10^{-13}$, for either gene being a complex member, and

OR = 4, $p<1\text{x}10^{-10}$ for both genes being members of the same complex, Fisher's exact tests). Overall our analyses suggest that 13–17% of the paralogs that are sometimes essential may have the variation in their essentiality associated with variable expression of their paralogs.

## Discussion

In this work, we have shown that paralogs are less frequently essential in cancer cell lines than singletons across a wide range of genetic backgrounds and that the frequency of a paralog's essentiality is influenced by its number of paralogs, their sequence similarity and their mode of duplication. In addition, we have shown that, for 13–17% of the paralogs that are variably essential in cancer cell lines, this variation can be attributed to paralog buffering. Specifically, the cell lines in which these genes are essential correspond to the cell lines in which at least one of their paralogs exhibits significantly lower expression, indicative of a synthetic lethal relationship. We found that these putative synthetic lethal relationships are more frequent among paralogs pairs that stem from whole-genome duplication as well as among paralog pairs that encode protein complex subunits.

### Limitations of CRISPR screens for estimating paralog essentiality in cancer cell lines

The definition of context-specific essentiality requires a reasonable estimation of which genes are and aren't essential in each context. In yeast, essentiality has been determined empirically based on the viability of strains with defined gene knockouts [1]. A similar resource is not available for human cells. In this work we applied a threshold to quantitative fitness scores derived from genome-wide pooled CRISPR screens in cancer cell lines in order to call each gene in each cell line either essential or non-essential. We note that genes identified as non-essential in our analysis may be essential under some other environmental conditions and this may only become evident as screens are performed in multiple conditions. The score threshold chosen was based on a data-driven analysis of the fitness score distribution across all cell lines (Fig 1B). However, as this threshold could potentially influence the downstream results, we also analyzed the data after binarizing the fitness scores with two additional thresholds: -0.4 and -0.6. Compared to the original threshold, these thresholds result in a larger and smaller set of sometimes essential genes, respectively, but all of the general trends that we observed regarding the relationship between essentiality and gene duplication hold (S3 Fig and S4 Fig for thresholds -0.4 and -0.6 respectively).

There are additional limitations to the use of extant CRISPR screens for analyzing the essentiality of genes, especially those with a highly sequence similar paralog. Most of the genes that were filtered out as a result of discarding multi-targeting sgRNAs are highly sequence-similar paralogs, primarily those with >90% sequence identity (S1B Fig). Our findings suggest that these genes are less likely to be broadly essential (Fig 3C) and more likely to exhibit a synthetic lethal relationship (Fig 6E) than paralogs with lower sequence identity. Excluding these genes may have had some influence on the results presented in this paper, but they represent a minority of all paralogs targeted (~4%), and consequently this influence is likely to be minor. To accurately assess the fitness consequences of these highly-sequence similar genes would require careful redesign of sgRNA libraries to avoid multi-targeting effects.

A further issue that could confound the estimates of cell line specific fitness effects is that cell line specific mutations could impact the efficacy of individual sgRNAs—e.g. single nucleotide variants may disrupt the match between the sgRNA protospacer and DNA sequence—or introduce additional multi-targeting effects [24]. The sgRNAs used in the screens analyzed were designed using the human reference genome and consequently may be subject to such

effects in the genetically variable set of cell lines under consideration. We expect that requiring a minimum of three different sgRNAs per gene should reduce the impact of genetic variation that alters the efficacy of individual guides or introduces off-target effects due to cell-line specific mutations. However, we cannot rule out that some of the observed context-dependent essentiality was influenced by bias in the sgRNA fitness scores resulting from cell-line specific mutations.

### Context-dependent essentiality and gene duplication

We find that, in cancer cell lines, context-specific essentiality is much more common than context-independent essentiality. In yeast it has been well established that different genes are essential in different environmental conditions [44], but there is also a growing recognition that gene essentiality is dependent on genetic background [7]. Understanding this dynamic view of essentiality can only be achieved using large-scale, genome-wide screens in multiple genetic contexts. In budding yeast and in the worm *C. elegans* such screens are starting to appear, but have been limited to only a couple of strains [10,11]. Here we have performed an analysis of 558 genetically heterogeneous cancer cell lines, allowing us to obtain a more comprehensive view of the variation in gene essentiality.

Analysis of the yeast knockout collection in a single genetic background previously suggested that WGDs were less likely to be essential than SSDs [21,22]. Here we expand this view to show that WGDs are less likely to be broadly essential than SSDs. However we also found that WGDs are less likely to be never essential than SSDs. This suggests that WGDs are more likely to be essential in a context-specific fashion than SSDs, an observation that can only be made using data from multiple genetic backgrounds. This increased context-specific essentiality of WGDs can be partially explained by the increased tendency of WGDs to display synthetic lethality—many WGDs appear to only be essential when the expression of their paralog is reduced. As cancer cell lines exhibit extensive genetic and epigenetic variability, the resulting variation in expression combined with the buffering relationships between paralogs appears to contribute to the variable essentiality of WGDs.

We estimate that at least 13–17% of cases of context-dependent essentiality among paralogs can be attributed to expression variation of a paralogous gene. We detected this by exploiting the transcriptomic variation in the cell line panel. There are likely additional buffering pairs that we did not detect because the genes were under-expressed in too few of the cell lines analyzed or because reduced expression (rather than no expression) is insufficient to reveal the synthetic lethal relationship. We note that genetic variation of the coding region of the paralogous gene, via copy number variation or nonsense mutation, is only observed in a minority of the putative synthetic lethal pairs (~36%). In the majority of cases the altered mRNA expression must be due to non-coding variation, epigenetic variation or more distal trans effects and remains to be explained.

There are a number of additional explanations for the ~85% of paralogs whose variable essentiality cannot be associated with expression variation of an individual paralog. In addition to buffering relationships between paralog pairs there are likely to be higher-order buffering relationships between multiple paralogs that form part of a paralog family. Such relationships might only be revealed when three or more paralogs are perturbed simultaneously. In budding yeast these higher-order interactions have been mapped systematically [45] but their contribution to the fitness of cancer cells remains unknown and we cannot easily detect them here. Furthermore, some of the sometimes essential paralogs may engage in synthetic lethal interactions with non-paralogous genes, i.e. genes with which they share no homology [46]. We do not have the statistical power to detect such relationships in an unbiased fashion, as it would

require comparing the variable essentiality of each paralog with the expression levels of all ~20,000 protein coding genes. Finally, it is likely that some of the variable essentiality of individual genes can be attributed to genetic variation that alters the gene itself rather than its paralogs. Such self-vs-self associations include oncogene addiction effects, where mutation or amplification of a gene in a tumor cell is associated with increased dependency upon that gene for cellular fitness, and 'CYCLOPS' effects, where single copy (heterozygous) gene loss results in increased sensitivity of cancer cells to further perturbation of the gene itself [47].

## Prioritizing paralog pairs most likely to exhibit synthetic lethal interactions

Synthetic lethality represents a promising approach for the development of targeted therapies in cancer, in particular synthetic lethal relationships involving genes that are recurrently mutated or deleted in cancer [48,49]. Many groups have sought to exploit redundancy between paralogs to identify new synthetic lethal drug targets in cancer [38,50–52]. However there is a large number of paralog pairs in the human genome and consequently an approach to identify those pairs most likely to be synthetic lethal would aid the identification of new synthetic lethal targets. Our findings suggest a simple approach to prioritize such paralog pairs by focusing on whole genome duplicates coding for subunits of known protein complexes. Many of the reported cancer synthetic lethalities fall into this category [33–37].

## Paralogs are a source of genetic robustness in tumor cells

Tumor genomes can harbor hundreds or thousands of genetic aberrations, ranging from missense mutations in individual genes to large scale losses of entire chromosomes [53,54]. That tumor cells remain viable, and indeed thrive, in the presence of these mutations suggests enormous robustness to genetic perturbations. Redundancy between paralogs has long been suggested as a means by which cells and organisms might achieve such genetic robustness [5,55,56]. Here we show that in tumor cell lines genetic perturbation of paralog genes is significantly better tolerated than perturbation of singleton genes, i.e. paralog genes are less likely to cause cell death when mutated. This suggests that buffering between paralogs may act as a source of genetic robustness in tumors, consistent with recent work analyzing mutational patterns in tumor genomes that demonstrated that paralog genes appear to be under weaker negative selection than singleton genes [57].

## Extending observations from budding yeast to understand essentiality in cancer cell lines

Overall, we found that many of the observations made using an individual strain of budding yeast can be extended to make predictions about the patterns of essentiality in human cancer cell lines. The observations about gene essentiality in budding yeast tend to be discussed in absolute terms, e.g. fewer paralogs than singletons are essential, due to the limited genetic context. Looking across a collection of genetically heterogeneous cancer cell lines we can extend these observations to an essentiality spectrum; e.g. we could extend the observation about essentiality to: paralogs are less frequently essential than singletons, i.e. essential in fewer genetic contexts. Similarly, we could extend the observation that WGDs are less likely to be essential than SSDs to the finding that WGDs are more variably essential—broadening the picture of WGD essentiality. Finally, synthetic lethal interactions observed between paralog pairs in budding yeast prompted us to investigate whether paralog buffering relationships could in part explain variable essentiality in cancer cell lines. Our ability to confirm and then build on

findings pertaining to gene essentiality from budding yeast indicates that studies of this model organism present a valuable source of predictions about essentiality in human cancer cell lines.

## Methods

Unless otherwise stated, analysis was performed with Python 3.6.5, Pandas 0.23 [58], SciPy 1.1.0[59], StatsModels 0.9.0[60] and scikit-learn 0.19.1[61].

### CRISPR data

We obtained the sgRNA-level raw log-fold change data for CRISPR screens in 558 heterogenous cancer cell lines from the Broad Institute's DepMap portal (release 19Q1, file: logfold_-change.csv). These screens were performed with 69,653 sgRNAs targeting 17,634 genes. From this dataset we dropped all guides with multiple on- or off-target alignments; this includes guides that matched perfectly to multiple locations in the genome, guides that matched with a single mismatch anywhere in the sgRNA sequence, and guides that matched with a double mismatch in the two most PAM-distal nucleotides (see sgRNA sequence alignments for how we identified these guides). We additionally filtered out guides that map to one genomic location but multiple genes, such as read-through genes, resulting in a total of 3,635 dropped guides. We processed the filtered log-fold changes file with CERES [12] to obtain gene-level scores. As a result of guide-filtering, 365 genes were no longer targeted by any guide, while 631 genes were left with only one or two guides targeting them. We deemed 1–2 guides insufficient for assigning a reliable score to a gene and thus dropped those genes from further analysis; leaving 16,638 genes that are targeted by at least 3 guides. We performed a precision-recall analysis of common essential genes [28] to compare our scores for these genes to those published in the DepMap portal (release 19Q1, file: gene_effect.csv).

### sgRNA sequence alignments

To identify multi-target sgRNAs, we used bowtie (version 1.2.2) [62] to align all sgRNA sequences against the hg19 reference genome, and allowing up to two mismatches (bowtie arguments -a -v 2). We used SAMtools (version 1.9)[63] to convert the SAM files output by bowtie to BAM files and then used Rsamtools (Bioconductor version 3.8)[64] to read the alignment data from the BAM file. We calculated the protospacer adjacent motif (PAM) start and end positions for each alignment and used the BSgenome R package (Bioconductor version 3.8)[65] to retrieve the sequences and dropped alignments that did not have the canonical NGG PAM. To derive a sgRNA to gene mapping, we matched genomic loci to genes based on hg19 gene annotations provided by the Consensus Coding Sequence project, using the GenomicRanges R package (Bioconductor version 3.8)[66] to find the overlapping regions. Our guide-gene map is similar to the one provided in the DepMap portal (release 19Q1, file: guide_gene_map.csv), with the exception that ours includes read-through genes.

### Binarizing fitness scores

Each gene in each cell line was assigned a fitness defect score by CERES but as we were interested in whether or not a given gene is essential in a given cell line in absolute terms, we converted the (continuous) scores to essential/non-essential calls. To achieve this we used scikit-learn's GaussianMixture class[61] to fit Gaussian mixture models (GMMs) with 1–5 components to the distribution of CERES scores and to determine the best fit model based on AIC score; this was a GMM with three components. The three component distributions of this model roughly correspond to: scores for non-essential genes, scores for genes that cause some

fitness defect but are not strictly associated with cell lethality, and scores for essential genes. We used the boundary point between the latter two categories, which is approximately -0.47, as the threshold score for essentiality: a given gene that scores at or below this threshold in a given cell line was considered essential in that cell line (Fig 1B).

### Gene expression data

From the DepMap portal (release 19Q1) we obtained gene expression data for 554 of the 558 cell lines that were used for the CRISPR screens (file: CCLE_depMap_19Q1_TPM.csv) [67,68].

### Gene copy number data

From the DepMap portal (release 19Q1) we obtained gene copy number data for 557 of the 558 cell lines that were used for the CRISPR screens, including the 554 cell lines for which gene expression data is also available (file: public_19Q1_gene_cn.csv).

### Nonsense mutations analysis

From the DepMap portal (release 19Q1) we obtained gene level mutation calls for the cell lines that were used for the CRISPR screens (file: depmap_19Q1_mutation_calls_v2.csv). To test whether the observed synthetic lethal effect for a paralog pair (A1, A2) could be explained by nonsense mutations in A2, we analyzed the mutations classified as 'Nonsense_Mutation' (column: Variant_Classification). For pairs whose A2 gene had a nonsense mutation in at least one cell line, we used Fisher's exact test to test whether there was an enrichment of A2 nonsense mutations in cell lines where A1 is essential compared to cell lines where A1 is not essential.

### Gene ID and symbol mapping

As different data sources used different gene IDs, we mapped genes to each other based on the HGNC [29] symbols report (Total Approved Symbols). This report was also used to identify read-through genes and protein-coding genes.

### Paralog data

Paralog relationships in the hg38 reference genome were obtained from ENSEMBL (release 93, reference genome grch38.p12) [26]. We retrieved the protein sequence identity for each paralog pair, calculated in both directions, e.g. for a paralog pair consisting of genes A1 and A2 we got the percent of A2's protein sequence matched in A1's protein sequence, and vice versa. The sequence identity is potentially asymmetrical due to the differing sequence lengths of the two genes. We restricted our set of paralog pairs to those which share 20% or more sequence identity in both directions, and where both genes are classified as protein-coding in the HGNC [29]. The paralogy information for each individual gene was summarized as the number of paralog pairs it is included in and the maximum percent of the gene's protein sequence that is matched in any of its paralogs. Genes were considered singletons if they did not appear in this summary.

### Whole genome vs. small-scale duplicates

We classified the paralog pairs in our dataset as whole genome duplicates (WGDs) if they were included on either of two lists of WGD pairs: the pairs identified by [31] and the high-confidence (strict) list identified by [32]. All remaining paralog pairs in our dataset, i.e. which were

not included on either of those lists, were classified as small-scale duplicates (SSDs). Individual paralog genes were marked as WGD if they were part of any WGD pair, regardless of whether they were also part of an SSD pair.

## Gene ontology enrichment analysis

We used gProfiler [69] to perform protein complex enrichment analysis of our set of broadly essential genes against our full set of protein-coding genes. For this purpose we input our list of 16,540 genes as the custom background (statistical scope) to test against and used the CORUM protein complex database as the data source [41].

## Supporting information

**S1 Table. CERES-corrected CRISPR scores for each protein-coding gene in every screened cell line, after filtering out guides that target multiple loci and/or multiple genes.**
(ZIP)

**S2 Table. Binary (essential/non-essential) calls for each gene in each cell line, i.e. the binarized version of S1 Table.**
(ZIP)

**S3 Table. Gene-level summaries: for each gene, the percent of cell lines in which it is essential, which essentiality category it falls in (never, sometimes or broadly), how many paralogs it has, the sequence identity percentage shared with its closest paralog (if any) and, whether it was generated through whole genome or small-scale duplication (if applicable).**
(CSV)

**S4 Table. Putative synthetic lethal paralog pairs identified among the sometimes essential genes with their most sequence-similar paralog, along with: the raw *t*-test and FDR corrected p-values for differential expression, their protein sequence identity percentage, their mode of duplication, whether either paralog is a member of a protein complex, whether both paralogs are members of the same protein complex, and whether the synthetic lethality effect appears driven by copy number variation or nonsense mutations.**
(CSV)

**S5 Table. Same as S4 Table but for the putative synthetic lethal pairs identified from the full list of paralog pairs that involve a sometimes essential gene.**
(CSV)

**S1 Fig. Establishing gene essentiality across 558 cancer cell lines.** (A) Bar chart showing the percentage of genes that are targeted by at least one multi-target sgRNA for all genes targeted in the CRISPR screens binned according to the protein sequence identity they share with their closest paralog (if any). (B) Bar chart showing the number of genes that were filtered out (dropped) among all genes binned according to the protein sequence identity the genes share with their closest paralog (if any). (C) Histogram showing the distribution of the count of genes that are essential across all cell lines.
(PDF)

**S2 Fig. The relationship between WGD, protein complex membership and synthetic lethality for all paralog pairs.** (A) Bar chart showing the percentage of WGD and SSD pairs in our dataset where at least one gene is a protein complex member. (B) Scatterplot of all tested pairs showing the mean expression of A2 in the cell lines where A1 is not essential (x-axis) or essential (y-axis). Colored dots indicate pairs for which the expression of A2 is significantly lower in

the cell lines where A1 is essential, i.e. putative synthetic lethal pairs, while grey dots represent pairs that were tested but not found to be synthetic lethal. Color corresponds to duplication type: pink for WGDs and green for SSDs. Selected synthetic lethal pairs involving protein complex subunits are labelled. Similar to Fig 5B, but here all sometimes essential paralogs pairs are included, instead of just the most sequence-similar pairs. (C) Bar chart showing the percentage of WGD and SSD pairs that are synthetic lethal. Similar to Fig 6A but here all sometimes essential paralog pairs are included. (D) Bar chart showing the percentage of synthetic lethal pairs among pairs where at least one gene is a protein complex member versus among pairs where neither gene is a protein complex member. Similar to Fig 6B but here all sometimes essential paralog pairs are included.
(PDF)

**S3 Fig. Variation in gene essentiality for fitness data binarized with score threshold -0.4.**
(A) Distribution of the percentage of cell lines in which a gene is essential when using a fitness score of -0.4 as the threshold for essentiality. The colors indicate the three broad categories of essentiality: genes that are essential in 0% of cell lines (blue), genes that are essential in more than 0 but at most 90% of cell lines (gray), and genes that are essential in 90% or more of the cell lines (orange). (B) Stacked bar graph showing the proportion of genes in each of the three essentiality categories that are paralogs (yellow) vs. those that are singletons (cyan). Genes that are never essential are significantly enriched in paralogs (OR = 1.5, $p < 2x10^{-16}$, Fisher's exact test) and genes that are broadly essential are significantly enriched in singletons (OR = 6.3, $p < 2x10^{-16}$, Fisher's exact test). (C) Stacked bar graph showing, for all genes binned according to the percentage of cell lines in which they are essential, the proportion of genes in each bin that are paralogs (yellow), or singletons (cyan). For reference, on the right is a bar showing the proportion of genes in the full dataset that are paralogs or singletons. (D) Stacked bar graph showing, for genes binned according to the number of paralogs they have, the proportion of genes in each bin that are never, sometimes and broadly essential. This proportion is significantly related to the number of paralogs ($p < 2x10^{-16}$, chi-squared test). (E) The kernel density estimates of the percent of a gene's protein sequence that is identical in its closest paralog, for paralogs in each essentiality category. The median sequence identity for never and sometimes essential genes (~48% and ~47.6% respectively) is significantly higher than the median sequence identity for broadly essential genes (~42.1%, $p = 0.002$, Mann-Whitney U test). Plot is drawn using Seaborn's kdeplot function with the default parameters. (F) Stacked bar graph showing the proportions of WGDs, SSDs and singletons that are never, sometimes and broadly essential. WGDs are more variably essential than SSDs: compared to SSDs, WGDs are enriched in sometimes essential genes (OR = 1.4, $p < 2x10^{-16}$, Fisher's exact test) and depleted in both broadly (OR = 0.37, $p = 1.2x10^{-14}$, Fisher's exact test) and never essential genes (OR = 0.78, $p = 4.3x10^{-9}$, Fisher's exact test).
(PDF)

**S4 Fig. Variation in gene essentiality for fitness data binarized with score threshold -0.6.**
(A) Distribution of the percentage of cell lines in which a gene is essential when using a fitness score of -0.6 as the threshold for essentiality. The colors indicate the three broad categories of essentiality: genes that are essential in 0% of cell lines (blue), genes that are essential in more than 0 but at most 90% of cell lines (gray), and genes that are essential in 90% or more of the cell lines (orange). (B) Stacked bar graph showing the proportion of genes in each of the three essentiality categories that are paralogs (yellow) vs. those that are singletons (cyan). Genes that are never essential are significantly enriched in paralogs (OR = 2.1, $p < 2x10^{-16}$, Fisher's exact test) and genes that are broadly essential are significantly enriched in singletons (OR = 5.8, $p < 2x10^{-16}$, Fisher's exact test). (C) Stacked bar graph showing, for all genes binned according

to the percentage of cell lines in which they are essential, the proportion of genes in each bin that are paralogs (yellow), or singletons (cyan). For reference, on the right is a bar showing the proportion of genes in the full dataset that are paralogs or singletons. (D) Stacked bar graph showing, for genes binned according to the number of paralogs they have, the proportion of genes in each bin that are never, sometimes and broadly essential. This proportion is significantly related to the number of paralogs ($p < 2 \times 10^{-16}$, chi-squared test). (E) The kernel density estimates of the percent of a gene's protein sequence that is identical in its closest paralog, for paralogs in each essentiality category. The median sequence identity for never and sometimes essential genes (~47.8% and ~47.5% respectively) is significantly higher than the median sequence identity for broadly essential genes (~41.3%, $p = 0.01$, Mann-Whitney U test). Plot is drawn using Seaborn's kdeplot function with the default parameters. (F) Stacked bar graph showing the proportions of WGDs, SSDs and singletons that are never, sometimes and broadly essential. WGDs are more variably essential than SSDs: compared to SSDs, WGDs are enriched in sometimes essential genes (OR = 1.35, $p = 2.5 \times 10^{-10}$, Fisher's exact test) and depleted in both broadly (OR = 0.39, $p = 1.3 \times 10^{-9}$, Fisher's exact test) and never essential genes (OR = 0.82, $p < 1 \times 10^{-5}$, Fisher's exact test).
(PDF)

## Acknowledgments

We thank Dr. Ariane Watson and Prof. Jonathan Bond for careful reading of the manuscript.

## Author Contributions

**Conceptualization:** Colm J. Ryan.

**Data curation:** Barbara De Kegel, Colm J. Ryan.

**Formal analysis:** Barbara De Kegel.

**Funding acquisition:** Colm J. Ryan.

**Investigation:** Barbara De Kegel.

**Methodology:** Barbara De Kegel, Colm J. Ryan.

**Supervision:** Colm J. Ryan.

**Visualization:** Barbara De Kegel.

**Writing – original draft:** Barbara De Kegel, Colm J. Ryan.

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
