## [Decision Letter · Decision Letter 0]

3 Sep 2019

Dear Dr Ryan,

Thank you very much for submitting your Research Article entitled 'Paralog buffering contributes to the variable essentiality of genes in cancer cell lines' to PLOS Genetics. Your manuscript was fully evaluated at the editorial level and by independent peer reviewers. The reviewers appreciated the attention to an important topic but identified some aspects of the manuscript that should be improved.

We therefore ask you to modify the manuscript according to the review recommendations before we can consider your manuscript for acceptance. Your revisions should address the specific points made by each reviewer.

[LINK]

Yours sincerely,

Aimee M. Dudley, Ph.D.

Associate Editor

PLOS Genetics

Gregory P. Copenhaver

Editor-in-Chief

PLOS Genetics

The reviewers appreciated the work and the manuscript itself, but had several questions and suggestions that should be addressed. One major concern, which I also had after reading the manuscript, is that (as the authors know) cancer cell lines are well known to harbor extensive aneuploidy and copy number variation. How much of the background-specific non-essentiality could be explained by the presence of multiple copies of the gene or its paralog(s)?

Other reviewer comments of particular importance are:

- the need to provide more detail about how fitness is calculated

- questions/ suggestions about the statistical methods used (continuous versus discretized data and t-tests versus a regression model)

Reviewer's Responses to Questions

**Comments to the Authors:**

Reviewer #1: In this paper, De Kegel & Ryan investigate the contribution of paralogs to variation in gene essentiality across human cell lines. The authors re-analyze the DepMap dataset of CRISPR-Cas9 KO screens in >500 cancer cell lines to explore this question. Similar to previous studies in model organisms, they find that duplicated genes are less likely to be essential than singleton genes. They extend this finding by showing that duplicated genes also show more variation in essentiality across many different genetic backgrounds, something that has not yet been feasible to investigate in model organisms. They further investigate the underlying mechanisms driving the essentiality of genes with paralogs, such as the number of paralogs a gene has and the mode of duplication by which the paralogs occurred.

The paper is well-written, well-structured, and uses clear and precise language. All relevant data is included in the supplementary tables. The authors frequently refer to previous work done in other organisms, and - to my knowledge - cite all relevant literature. They clearly describe the used methods and have performed a thorough cleaning of the DepMap dataset before analysis.

While many of the findings in the article are intuitive given previous knowledge from model organisms, these observations have not previously been described in human cells. In addition, the authors take their work a step further than previous studies by looking at variability in gene essentiality among many different genetic backgrounds. This paper is thus certainly useful to the scientific community. However, I have a few comments, mainly about the detection and definition of differential essential genes (“sometimes essential genes”), which may affect all further downstream analyses.

Main comments:

1. Page 6/7 - “~86% of the genes that share 90% or more sequence identity with their closest paralog are targeted by at least one multi-target sgRNA in the Avana library”. Are there still 3 or more gRNAs remaining for these genes in the dataset, or have most of the highly similar paralogs been filtered out? If many or most of the highly similar genes have been removed, can the authors comment on how this affects the results?

2. Page 7/8 – The authors describe how they use a single CRISPR score threshold (-0.47) to determine gene essentiality. This means that genes with an average score close to this cutoff would be classified as differential essential even though the actual scores may be very comparable across cell lines (for example -0.45 in some cell lines and -0.49 in others). In addition, genes with a score just below -0.47 in one cell line and above -0.47 in all others would be classified as differential essential. This could be prevented by for example excluding all scores in the -0.45 to -0.50 range. Could the authors comment and/or show data on the effect of their choice of this single threshold on the set of differential essential genes?

3. Page 6-8 - If a given cell line carries a mutation (compared to the reference genome) in the sequence targeted by a guide, this guide may be less efficient in this cell line. Could the authors comment on whether some of the observed differential essentiality could be caused by differences in guide efficiency due to mutations or other sequence differences between cell lines?

Minor comments:

1. Page 13/14 - The authors use expression data to identify gene pairs where the essentiality of a gene A correlates with lower expression of its paralog B. I agree with the authors that these could indicate potential synthetic lethal relationships, but other explanations are also possible. Although at several places they describe these as “putative” synthetic lethal relationships, at other moments they talk in more definitive terms about synthetic lethal pairs. In my opinion, consistent use of more careful phrasing like “putative synthetic lethal pairs” would be preferable.

Reviewer #2: See uploaded attachment

Reviewer #3: In this study, De Kegel and Ryan examine the set of genes required for viability across 558 tumor cell lines, as determined by the DepMap project at the Broad institute. The authors investigate why certain genes are essential in all, some or none of the tested cell lines and to what extent this variability can be explained by gene duplication/paralogy. The authors report that, while paralogs are overall less likely to be essential, the ones that do have an essential phenotype in at least one context provide insight into the mechanisms of variable essentiality. Specifically, the probability of a gene being required for viability depends on the number of its cognate paralogs, their mutual sequence similarity and relative expression levels, as well as the nature of the duplication event they originated from.

I found this study to be very well designed, executed and described. The question posed is very interesting and takes great advantage of an extensive body of work in model organisms (i.e., yeast) as well as the latest technologies that have recently generated relevant datasets for human cancer cell lines. The analysis is thorough and rigorous, and the description of the results and interpretations is very clear and precise. I think this would be an important contribution to the field of functional genomics and cancer biology.

I don't have any major comments or suggestions on the manuscript.

A few minor comments are:

- The authors estimate that 13-17% of paralogs that are essential in some, but not all, cell lines are explained by their genetic dependence on a paralog partner that's variably expressed across the cell lines. Could the authors provide their intuition/speculation/preliminary insight into what could explain the other (more substantial) fraction of variably essential paralogs? Could it be synthetic lethality with other non-paralog genes that are also variably expressed across the cell lines? And if the genome-wide expression data for these cell lines is available, could one attempt to identify these synthetic lethal partners based on the correlation of their expression levels with fitness levels?

- Maybe I missed it throughout the text and the methods, but could the authors clarify where is their synthetic lethality data coming from? E.g., lines 290-294: "we found that among WGD paralog pairs, ~16.5% are synthetic lethal" -- what dataset was used to perform this assessment?

- Also, lines 303-305: "Approximately 19.1% of the paralog pairs consisting of at least one protein complex member gene are synthetic lethal, compared to only ~9.3% of the non-protein-complex pairs" -- what is the overall frequency of synthetic lethality among complex- and non-complex pairs (regardless of paralogy)?

**Have all data underlying the figures and results presented in the manuscript been provided?**

Reviewer #1: Yes

Reviewer #2: Yes

Reviewer #3: Yes

PLOS authors have the option to publish the peer review history of their article (what does this mean?). If published, this will include your full peer review and any attached files.

Reviewer #1: No

Reviewer #2: No

Reviewer #3: No

---

## [Editor Report · Decision Letter 1]

8 Oct 2019

Dear Dr Ryan,

We are pleased to inform you that your manuscript entitled "Paralog buffering contributes to the variable essentiality of genes in cancer cell lines" has been editorially accepted for publication in PLOS Genetics. Congratulations!

Yours sincerely,

Aimee M. Dudley, Ph.D.

Associate Editor

PLOS Genetics

Gregory P. Copenhaver

Editor-in-Chief

PLOS Genetics

Comments from the reviewers (if applicable):

I have read the detailed response and believe that the authors have thoroughly addressed the concerns of the 3 reviewers. The revision is acceptable for publication in PLoS Genetics.

**Data Deposition**

http://datadryad.org/submit?journalID=pgenetics&manu=PGENETICS-D-19-01244R1

**Press Queries**

---

## [Editor Report · Acceptance letter]

21 Oct 2019

PGENETICS-D-19-01244R1 

Paralog buffering contributes to the variable essentiality of genes in cancer cell lines 

Dear Dr Ryan, 

We are pleased to inform you that your manuscript entitled "Paralog buffering contributes to the variable essentiality of genes in cancer cell lines" has been formally accepted for publication in PLOS Genetics! Your manuscript is now with our production department and you will be notified of the publication date in due course.

With kind regards,

Kaitlin Butler

PLOS Genetics

On behalf of:
